# How pharmacy and medicine students experience the power differential between professions: "Even if the pharmacist knows better, the doctor's decision goes"

Josephine Thomas[1,2]*, Koshila Kumar[3], Anna Chur-Hansen[1]

**1** School of Psychology, Faculty of Health and Medical Sciences, University of Adelaide, Adelaide, South Australia, Australia, **2** Adelaide Medical School, Faculty of Health and Medical Sciences, University of Adelaide, Adelaide, South Australia, Australia, **3** Prideaux Centre for Research in Health Professions Education, Flinders University, Bedford Park, South Australia, Australia

* josephine.thomas@adelaide.edu.au

**Data Availability Statement:** Data cannot be shared publicly because the research participants for this study did not provide consent for data

## Abstract

Interprofessional Education (IPE) is one approach to improving communication and collaborative practice between professions, which are essential for the optimal delivery of healthcare. Common barriers include negative attitudes, professional stereotypes, professional cultures and power differentials between professional groups. The aim of this qualitative study was to explore how professional hierarchies and power differentials shape interprofessional interactions between preregistration pharmacy and medicine students. Data were gathered via semi-structured interviews and subject to thematic analysis. Four main themes were identified: Reproducing traditional hierarchies; Social norms around respect; Hierarchies in care values and goals; and Challenging the narrative is possible. Students' interactions with and views of the other profession largely reflected traditional stereotypes and power differentials. Hierarchy was evident in how respect was accorded and in how care values and goals were managed. Despite this, students overwhelmingly perceived and reported a sense of agency in changing the status quo. Emerging professional identity and conceptualisation of future roles is heavily influenced by the hierarchical relationship between the professions and can pose a significant barrier to collaborative practice. Greater support for collaborative interprofessional practice is needed at the level of policy and accreditation in health education and healthcare to ensure greater commitment to change.

## Introduction

There in an unprecedented increase in the complexity of healthcare provision. In the prescribing context, clinicians increasingly encounter multimorbid patients with polypharmacy and a high likelihood of drug interactions that increase the potential for harm [1, 2]. In this context, a collaborative working relationship between doctors and pharmacists in essential to facilitate optimal integration of professional expertise and to deliver safe, high-quality healthcare. The

sharing. Unfortunately, it is not possible to retrospectively obtain consent to allow us to share the data publicly. The University of Adelaide, School of Psychology, Human Research Ethics Subcommittee and Flinders University Social and Behavioural Research Ethics Committee have approved the research protocol with the specified conditions: -The data will only be accessible to the researchers. -It is anticipated that the results from this study will be published in a peer reviewed article. No participants will be identified in the publication and only aggregated data will be published. Data are available from the University of Adelaide, School of Psychology, Human Research Ethics Subcommittee: ph 61 8 8313 5693, fax 61 8 8313 3770, email: psychologyoffice@adelaide.edu.au; for researchers who meet the criteria for access to confidential data. (approval no. 16/19).

**Funding:** The author(s) received no specific funding for this work.

**Competing interests:** The authors have declared that no competing interests exist.

involvement of both pharmacist and medical practitioner in the prescribing process, has the potential to increase patient safety through an additional layer of cross-checking [3, 4].

However, collaborative practice can be hampered by negative attitudes, professional stereotypes, professional cultures and power differentials between professional groups [5, 6]. In the prescribing context, pharmacists are reluctant to question a medical practitioner's authority [7]. Pharmacists' reticence may be partly due to the traditional unequal power relationship between the professions [5, 8]. Historically, medical dominance has been a societal norm afforded by perceptions of doctors as saving and prolonging lives [9, 10]. Medical dominance is upheld by legal sanctions and demarcations, such as the legislative requirements of prescribing, where the doctor holds the ultimate power of medication choice and legal responsibility for care [11, 12]. The notion of doctor as leader is embedded in medical professional identity [5], and further reinforced by the patriarchal doctor-patient relationship. [13] and the evolution of medicine by exclusion of other professions [14]. The sociological literature demonstrates that medicine is a privileged professional group, exercising patriarchal control [15, 16].

Interprofessional education (IPE) is a common mechanism for promoting collaborative practice including with students, with the hope this will translate into future practice [17]. However, interprofessional education alone is not sufficient to create collaborative practitioners in the context of embedded hierarchies within the interprofessional healthcare team [18]. Attitudes and behaviours in the clinical environment and in society more broadly also impacts students' interactions with other professionals [19–21].

The power differential between professions is an important consideration in planning, implementing and evaluating IPE. Theoretical frameworks underpinning IPE suggest that conditions to reduce prejudice in the context of contact between different groups include equal status [22–24]. There are relatively few studies which directly address the issue of power relationships in IPE. An analysis of published IPE research spanning several decades, identified only 6 articles which addressed power and conflict [25]. Where the issue of power is explicitly addressed, the doctor is consistently articulated as the dominant professional [26]. An earlier study by the current authors, showed that medicine and pharmacy students' interactions reflected traditional power differences and professional hierarchies between the professions, with the medical prescriber ascribed greater responsibility, and the pharmacist a supporting role [27].

The aim of this study was to explore how professional hierarchies and power differentials shape interprofessional interactions and learning between pharmacy and medicine students. We posed the research question: How do traditional professional hierarchies influence preregistration medicine and pharmacy students' interactions and learning with, from, and about each other?

## Methods

### Methodology/Research design

A qualitative design was used to explore students' experience of how the power differential between medicine and pharmacy impact on their interprofessional interactions.

### Participants and context

Participants were preregistration pharmacy and medical students from two universities in Australia. The medical student cohort comprised 198 students who completed Year 4 of a 6-year program and the pharmacy cohort comprised 114 students who completed Year 3 of a 4-year program, in the year prior to this study. Both groups had commenced clinical (work based) placements in the year prior to this study, including participation in interprofessional

workshops. The workshops involved small groups of pharmacy and medicine students (2–3 students of each profession) working together to solve clinical problems for case-based scenarios. Tasks included: prescribing for the case; determining doses; predicting adverse effects and role-play of consultations. Educators from both professions acted as roving tutors and co-presented a summary wrap-up session.

## Data collection

Students were invited via their university student email to participate in a semi-structured interview about their experiences of interacting with another professional group in the context of the interprofessional workshops and more broadly, in their clinical placements. Participation in the research was voluntary and there was no impact on grades. Nine medicine and seven pharmacy students gave informed consent for interview. All interviews were audiotaped, transcribed and assigned a unique ID number. Interviews were conducted by two personnel who were not involved with teaching or assessment of the participants. Both interviewers used pre-determined prompt questions and met with the first author to discuss alignment of interviews with the research focus. The interviewers did not participate in the design or analysis of the study.

## Data analysis

Data were analysed using a thematic analysis approach [28, 29]. The analysis involved a process of detailed coding during which codes were identified. The second and third authors reviewed the first author's preliminary coding to cross check and develop consensus. The codes were grouped into subthemes and themes. Themes and subthemes were listed in a matrix with illustrative quotes from individual participants for each code.

## Ethical considerations

Ethics approval was granted by the relevant institutional Ethics Committees The University of Adelaide, School of Psychology, Human Research Ethics Subcommittee (approval 16/19); and Flinders University Social and Behavioural Research Ethics Committee (approval OH-00087). Eligible students were provided with an information sheet outlining the study and written consent was obtained from all consenting participants.

# Results/Findings

Four main themes were identified: Reproducing traditional hierarchies; Social norms around respect; Hierarchies in care values and goals; and Challenging the narrative. Illustrative quotes are presented for each theme (Participant ID: M = medicine, P = pharmacy).

## Reproducing traditional hierarchy

This theme incorporates students' perceptions of being socialised into identities that reflect a power differential between medicine and pharmacy professions, both within academic and clinical learning contexts. Pharmacy students viewed that the subordinate role of the pharmacist was reinforced and perpetuated in their university teaching and training. They noted that non-doctors are typically expected to demonstrate deference and not to question the judgements of doctors.

> "*I guess we have it drilled into us that the doctor is the font of all knowledge and we should defer to them on any matters we are not sure of. . . . . . we have that subordinate role drilled*

*into us where it's just suggest [to the doctor] 'maybe you would like to do this, what do you think?" P7*

This subordinate identity influenced how pharmacists communicated with doctors and doctors in training. Pharmacy participants reported they felt uncomfortable to voice their opinion and unable to engage in a discussion as equals. They noted that despite their medication and prescribing credentials, medical students and doctors were not always accepting of their input.

"*the pharmacist has the responsibility to call and inform the doctor but sometimes it is really hard for the doctor to accept" P6*

They viewed that the medical students were over-confident in their medication decision making, despite an obvious lack of underpinning pharmacotherapeutics knowledge and rationale.

"*and then we realized they didn't know the difference between like the potencies of the statins and that sort of thing . . ..." P3.*

In turn, medical students viewed pharmacy students as unable to assert their opinions and as timid and unskilled in communication. This seemed to provide medical students with a rationale for forging ahead and taking leadership in the decision making.

"*but then I also noticed that that same [pharmacy]student was really reluctant to make a choice, sometimes it felt like in cases like that OK it felt a bit obvious what was the choice, but they were like 'Oh no, at the end of the day you guys will have to make the decision'. I don't know, I thought like they knew a lot more but then there were reluctant to like they could have taken the lead, I felt"M3.*

Most medical students explicitly denied the existence of power gradients. However, the way they talked about the doctor as the leader of the healthcare team and the subordinate nature of the interaction, contradicted this.

"*The allied health professional kind of comes in and does an assessment and makes recommendations but I don't think anything really gets instigated, I don't think until the doctor agrees and says its ok" M2*

From the medical student perspective, doctors had ultimate legal responsibility for decisions on care, so their role was to function as arbiters of different viewpoints. However, some medical participants acknowledged that embedded hierarchies and entrenched ways of practice could mean that the doctor could potentially disregard sound advice.

"*And so, even if the pharmacist knows better, the doctor's decision goes, and the patient may potentially have a worse outcome" M2*

## Social norms around respect

This theme incorporates students' perceptions of respect and who was worthy of respect, which appeared to influence their own attitudes toward and level of engagement with the

other group. For pharmacy students, their professional identity and value was built around medication expertise and this added to their confidence and sense of self.

"*and you sort of reconfirmed your position as a pharmacist, for me anyway, just like they really do look to us for that clarity and that knowledge on anything related to medicines*" P2.

However, they perceived medical students as careless and overconfident in the prescribing role and lacking in respect for pharmacy perspectives.

"*there was just a bit more sort of talking over the top and we would give out a recommendation and they would sort of ignore it and then when we got the answers for the case the pharmacy students were right anyway*" P2

Pharmacy students noted that clinically oriented skills and activities, such as medical diagnosis were generally more highly regarded and respected, and this was more pronounced in clinical settings in the context of a patriarchal doctor-patient relationship [13].

"*From my personal interaction I believe that to be true and even I think for being a sort of awareness that doctors are held to higher regard than pharmacists even though we both have different strengths and different knowledge bases*" P2

Medical students were quick to describe their respect for individuals with expertise and knowledge. However, this was mostly medical profession centric and usually linked with position in the medical hierarchy e.g. the consultant (head of the specialist hospital medical team). In some instances, there was admiration and respect for other healthcare professionals, but this was more the exception than the norm. They regarded that expertise provided by the pharmacy students included detailed medication knowledge, particularly about adverse effects and drug interactions; and that was appreciated by both professional groups.

"*If you picked a drug, they would know the science and they would understand it, I thought that was an acute strength*" M7

Medical students considered that taking responsibility and leadership were integral aspects of the medical role and identity. They did not view this as being disrespectful behaviour toward pharmacy students and toward the pharmacy profession generally.

"*I think that position of responsibility and leadership, right or wrong and this model or another, often falls to the doctor just based on long standing societal roles*" M4

Some medical participants did recognise that some medical disciplines may be more likely to act in a superior way, with surgeons cited as the typical example. However, even in other disciplines with arguably more functional interprofessional teams (e.g. geriatric medicine, rehabilitation), doctors were still perceived as the team leaders with ultimate decision-making ability.

"*like I don't think any of us had the attitude of surgeons or anything just yet, like there is always a couple who think highly of [themselves]*" M6

## Hierarchies in care values and goals

This theme refers to the notion that students were developing awareness of which clinical values and goals mattered in healthcare provision. The main focus of the pharmacist was viewed as promoting medication safety, avoiding interactions and adverse effects. In contrast, the main focus of the doctor was perceived to be diagnosis and treatment of symptoms and disease. These distinctly different orientations to patient care, were seen to be in conflict at times, and pharmacy students identified they had to engage in convincing doctors about different possibilities to consider.

> "*if there is a need for some medicines to be changed, we do have to speak to the doctor and try to convince them so explain the benefits of changing and the pros and cons between each of them*" P4

Medical students regarded pharmacists as providing a safety net through "double checking" the doctors' prescriptions for potential medication adverse effects, dosing errors and interactions. However, pharmacy students regarded this further reinforced the lesser role of pharmacist.

> "*that's where the pharmacist comes in and sort of neatly tie up what the doctor has started*" P2

While medical students were aware of the safety role of a pharmacist, some regarded this as a hindrance to doctors doing their job.

> "*they always say to stop this because they have a headache because of this but they don't have a headache because of that drug, they had a headache well before they started the drug.*" M7

The lack of understanding about the differing healthcare goals and values between the two professions was regarded as a barrier to collaborative practice.

> "*they don't just understand the role of the pharmacist and what we actually do. . . . . . doctors might say 'What are you doing' or 'Why aren't you doing that', it's just a lack of awareness of each other*" P2

## Changing the narrative

This theme relates to students' perceived agency to change the narrative and challenge the status quo in the relationship between pharmacy and medicine. Both groups recognised that there was a diversity of attitudes within each profession and that some members may be more willing to listen and collaborate. Pharmacy students acknowledged a diversity in attitudes among medical students, reflecting a sense of hope that future medical professionals may be willing to take a different approach

> "*[there are] definitely medical students and future doctors out there who are really positive and really willing to take on a different outlook*" P2

Some medical students recognised the need to reflect and monitor behaviours, including their own, that are part of the hierarchical relationship between the professions.

"*being self-aware enough that you know, don't have a unique sense of invulnerability that "Oh I will never be disrespectful" or "I will never stop listening"*"M9

Both groups perceived that they had a responsibility to engage in gaining a better understanding of other professions' roles.

"*maybe spend five minutes talking with your pharmacy colleagues about what they are up to in their degree and what they have been learning in the last few months. . . which sounds really trite but I think in retrospect actually it is a useful think to know, to have that understanding*" M4

Improving interprofessional relations was seen as a way to bridge the professional divide that currently existed between the professions.

"*what I see that doctors are the main pillar and we are like the cement, pharmacists are like the cement like to help the patient you know, the stronger the pillar, the stronger the building is going to be*" P3

Formal interprofessional education sessions were regarded as being helpful for improving interprofessional interactions and attitudes toward other professions. However, there was recognition that interprofessional interaction in undergraduate learning is a limited part of the formal curriculum, and more of an incidental occurrence in clinical rotations.

"*because we will be working together later once we graduate and go into an actual work environment anyway so if we have prior exposure to it. I think it would be really helpful*" P1

## Discussion

The traditional professional roles and the power differential between professions are a strong influence on how preregistration medical and pharmacy students interact and learn with, from and about each other. Their perception of the roles within a healthcare team, was dominated by the power differential of the professional hierarchy. The emerging professional identity of both groups was strongly influenced by traditional stereotypes and socialisation in both educational and clinical settings. However, both groups recognised the potential negative impact of a hierarchical relationship between professions. However, some students seemed prepared to challenge the status quo.

Although the existence of a power gradient was explicitly denied by medical students, it was paradoxically apparent in their language and attitudes. They described themselves as "leaders", and assumed that role, even if they did not have the necessary medication knowledge or prescribing skills for the situation. This type of behaviour has been observed in other studies [30], with medical students and doctors compelled to work things out themselves, and to take charge despite gaps in experience and knowledge [30, 31]. The role of doctor as leader has implications in the context of the interprofessional team. It is well recognised that power gradients can facilitate an unsafe healthcare environment, and this applies both within, and across professions [32]. A distributed leadership model, where leadership is an action taken by the person with the best available skillset at that time, is more likely to be a successful model for utilisation of the available expertise of all team members [33]. It is unclear if medical students perceive their own leadership in this way, and there would be further benefits for future research to address student perceptions of models of leadership in an interprofessional team.

The medical students' attitude of superiority toward the pharmacy students seen in this study is something that is well described in the literature [34, 35]. This has been recognized as a likely effect of strong cohesion and identification with their professional group [36]. Although strong "in-group" identification has a positive impact on professional identity development, there are negative outcomes of this strong identification, including a lack of collaborative behaviours and prejudice [23]. Although medical students recognise the need for teamwork skills, they commonly demonstrate an exclusive attitude to learning with other healthcare students [35]. In this study the medical students did not explicitly see a power differential within their own interactions. However, they did refer to some disciplines (notably surgery), as being prone to behaving in a superior manner and some expressed the intention to avoid adopting these values and behaviours.

Professional hierarchy was evident to the pharmacy students and they commented on this explicitly, and with some resignation. The recognition of medical practitioners' professional standing is likely deeply held, beginning prior to entry to study for many. Influences may include family, particularly for medical students who are likely to have a family member within the profession [37, 38]. The intention of educators is to promote collaborative behaviours by bringing two professional groups together, in the setting of interprofessional learning [17]. However, the emerging professional identities of these two groups do begin on an equal footing. This inequality carries a risk of interprofessional contact reinforcing stereotypes and maintaining the dominance of medicine, rather than the intended outcome of promoting collaborative behaviour [23].

A predominant aspect of pharmacists' identity is related to the emphasis on scientific knowledge and calculation in their training, with these seen as desirable attributes, particularly in their interaction with doctors [39]. The nature of pharmacy practice is careful and exacting work; they follow algorithms and validated systems and tend to be risk averse [7]. This may account for their expressed view of medical students as careless. For the pharmacy students in this study, the detailed medication knowledge they possess was seen a source of professional strength, and a way to garner respect and gain doctors' attention in professional interactions. Medical students see knowledge as a commodity and view pharmacy students' knowledge as something they wish to use or acquire [27]. In this study medical students tended to equate knowledge with power and position in the hierarchy, and individuals with superior knowledge commanded respect. However, they were still willing to make therapeutic decisions without adequate knowledge, which appears to be contradictory, with their assumption of the superior role overriding this knowledge deficit, despite the inherent danger to patients.

Students demonstrated awareness of how power differentials and professional stereotypes can negatively influence collaborative practice. Some expressed an intention to address these issues in pursuit of more collaborative professional working relationships in the future.

Interprofessional education may be a conduit for developing positive collaborative relationships in practice. However, this requires educators to move beyond a focus on content, to reveal the established ways of thinking, seeing, and doing that can influence interprofessional interactions to help students to negotiate and manage these. Curricular strategies can be implemented to help students in recognising the impact of professional hierarchy on their behaviour. Methods that may be useful in this regard include expert facilitators engaging students in discourse about professional identity, power, hierarchy and setting the tone around respectful behaviours and ways of interacting. Debrief may be an effective model to improve self-awareness about interactions, since it is designed to build on experiential learning and can encompass affective components of learning [40, 41].

Institutional support for more collaborative practice has been provided at several levels across education, training and the healthcare workforce; and continues to provide support for

change [42–44]. However, the time spent on interprofessional learning remains small in comparison to other elements of curriculum and this type of learning is less likely to be assessed, diminishing its perceived importance [45]. Furthermore, students will continue to experience role modelling that perpetuates professional stereotypes in the clinical domain. Ongoing reform of the prescribing process should be considered to enable greater use of pharmacist expertise in medication safety. Pharmacist prescribing continues to gain support [46].

Greater commitment to change may be achieved through policy and accreditation requirements in health education and healthcare, that address the power balance between professions and promote collaborative practice. Interprofessional competencies should be an expected common outcome at completion of study, across all health professions. Mandating the demonstration of interprofessional competencies for health professions graduates through accreditation, would ensure that these competencies feature more prominently in learning and assessment of students. Models of interprofessional practice could be similarly promoted through standards of accreditation and continuing professional development for qualified health professionals.

Despite the small sample size, the data obtained from the interviews are rich and deep, offering a strong narrative around the research question. In addition, there was an element of data triangulation achieved from our previous research (which utilised students' written reflections), as well as triangulation of perspectives across multiple researchers. This adds to the credibility of the findings in line with the quality criteria for qualitative research [47].

## Conclusion

The traditional hierarchical relationship between the medicine and pharmacy professions poses a significant barrier to collaborative practice. Medicine and pharmacy students reported that they value learning about the other profession and appear prepared to challenge traditional roles and power differentials. We recommend educators facilitate discussions about professional identity, power and hierarchy in pharmacy and medical curricula. Greater institutional support for collaborative interprofessional practice is needed at the level of policy and accreditation in both health education and healthcare.

## Acknowledgments

Dr. Alice O'Connell, Dr. Paul Kleinig and A/Prof Michael Wiese for their assistance with workshops and interviews.

## Author Contributions

**Conceptualization:** Josephine Thomas, Koshila Kumar, Anna Chur-Hansen.

**Formal analysis:** Josephine Thomas, Koshila Kumar, Anna Chur-Hansen.

**Investigation:** Josephine Thomas.

**Methodology:** Josephine Thomas, Koshila Kumar, Anna Chur-Hansen.

**Supervision:** Koshila Kumar, Anna Chur-Hansen.

**Writing – original draft:** Josephine Thomas, Koshila Kumar, Anna Chur-Hansen.

**Writing – review & editing:** Josephine Thomas, Koshila Kumar, Anna Chur-Hansen.

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
