## [Decision Letter · Decision Letter 0]

7 Jul 2021

PONE-D-21-12355

How pharmacy and medicine students experience the power differential between professions: “Even if the pharmacist knows better, the doctor’s decision goes.”

PLOS ONE

Dear Dr. Thomas,

Thank you for submitting your manuscript to PLOS ONE. After careful consideration, we feel that it has merit but does not fully meet PLOS ONE’s publication criteria as it currently stands. Therefore, we invite you to submit a revised version of the manuscript that addresses the points raised during the review process.

We look forward to receiving your revised manuscript.

Kind regards,

Shaun Wen Huey Lee

Academic Editor

PLOS ONE

Journal Requirements:

2. Please include additional information regarding the interview guide used in the study and ensure that you have provided sufficient details that others could replicate the analyses. For instance, if you developed an interview guide as part of this study and it is not under a copyright more restrictive than CC-BY, please include a copy, in both the original language and English, as Supporting Information.

3. Please provide additional details regarding participant consent. In the ethics statement in the Methods and online submission information, please ensure that you have specified whether consent was informed.

Reviewers' comments:

Reviewer's Responses to Questions

**Comments to the Author**

1. Is the manuscript technically sound, and do the data support the conclusions?

Reviewer #1: Yes

Reviewer #2: Yes

2. Has the statistical analysis been performed appropriately and rigorously? 

Reviewer #1: N/A

Reviewer #2: No

3. Have the authors made all data underlying the findings in their manuscript fully available?

Reviewer #1: No

Reviewer #2: Yes

4. Is the manuscript presented in an intelligible fashion and written in standard English?

Reviewer #1: Yes

Reviewer #2: Yes

5. Review Comments to the Author

Reviewer #1: well planned and executed study.

how do you made sure that all the participants attended IPE workshops?

What's your recommendations to improve the collaboration between medicine and pharmacy professionals?

Reviewer #2: Review Comments to the Author

Please use the space provided to explain your answers to the questions above. You may also include additional comments for the author, including concerns about dual publication, research ethics, or publication ethics. (Please upload your review as an attachment if it exceeds 20,000 characters) (Limit 200 to 20000 Characters)

The sample size included in the study was very small. How would the author clarify this point?

6. PLOS authors have the option to publish the peer review history of their article (what does this mean?). If published, this will include your full peer review and any attached files.

Reviewer #1: No

Reviewer #2: No

---

## [Author Response · Author response to Decision Letter 0]

26 Jul 2021

Formatting has been revised to comply with journal requirements. Title page added. References have been checked.

2. Please include additional information regarding the interview guide used in the study and ensure that you have provided sufficient details that others could replicate the analyses. For instance, if you developed an interview guide as part of this study and it is not under a copyright more restrictive than CC-BY, please include a copy, in both the original language and English, as Supporting Information.

The interview prompt questions have now been included in the supporting information.

3. Please provide additional details regarding participant consent. In the ethics statement in the Methods and online submission information, please ensure that you have specified whether consent was informed.

Participants were provided with an information sheet and completed a written consent form (lines 132-133). This information has now been included in the manuscript, specifically the statement that “students gave informed consent for interview” (line 116).

Participant information sheet and consent form also provided in supporting information

4. We note that you have indicated that data from this study are available upon request. PLOS only allows data to be available upon request if there are legal or ethical restrictions on sharing data publicly. For more information on unacceptable data access restrictions, please see http://journals.plos.org/plosone/s/data-availability#loc-unacceptable-data-access-restrictions

The research participants for this study did not provide consent for data sharing. Unfortunately, it is not possible to retrospectively obtain consent to allow us to share the data publicly. The University of Adelaide, School of Psychology, Human Research Ethics Subcommittee AND the Flinders University Social and Behavioural Research Ethics Committee have approved the research protocol with the specified conditions:

-The data will only be accessible to the researchers.

-It is anticipated that the results from this study will be published in a peer reviewed article. No participants will be identified in the publication and only aggregated data will be published. 

Contact for The University of Adelaide, School of Psychology, Human Research Ethics Subcommittee: ph 61 8 8313 5693, fax 61 8 8313 3770, email: psychologyoffice@adelaide.edu.au

Full names of ethics committees provided in manuscript (at lines 129-131)

5. Reviewer #1: well planned and executed study.

how do you made sure that all the participants attended IPE workshops?

What's your recommendations to improve the collaboration between medicine and pharmacy professionals?

Non-attendance was an infrequent event. However, both the recruitment notice and the information sheet clearly outlined that attendance at workshops was a prerequisite for participation in the research activity. Additionally, the interview included an opening statement and question about the workshops. Provision of the interview questions and participant information sheet in the supporting information, clarifies this.

Our recommendations are for educators to facilitate discussion about professional identity, power, hierarchy in health professions curricula. Additional sentence added to conclusion. We have also advocated for greater institutional support through policy and accreditation, and further clarified this statement in the discussion (lines 381- 383 and 392-393).

Reviewer #2: Review Comments to the Author

The sample size included in the study was very small. How would the author clarify this point?

Despite the small sample size, the data obtained from the interviews are rich and deep, offering a strong narrative around the research question. In addition, there was an element of data triangulation achieved from our previous research (which utilised students’ written reflections), as well as triangulation of perspectives across multiple researchers. This adds to the credibility of the findings in line with the quality criteria for qualitative research.

We have included a paragraph at the end of the discussion section (lines 399-403

Ref. Frambach, J. M., van der Vleuten, C. P., & Durning, S. J. AM last page: Quality criteria in qualitative and quantitative research. Academic Medicine. 2013,88(4), 552.

---

## [Decision Letter · Decision Letter 1]

16 Aug 2021

How pharmacy and medicine students experience the power differential between professions: “Even if the pharmacist knows better, the doctor’s decision goes.”

PONE-D-21-12355R1

Dear Dr. Thomas,

We’re pleased to inform you that your manuscript has been judged scientifically suitable for publication and will be formally accepted for publication once it meets all outstanding technical requirements.

Kind regards,

Jenny Wilkinson, PhD

Academic Editor

PLOS ONE

Additional Editor Comments (optional):

Thank you for your revisions, both reviewers have now recommended acceptance of the work

Reviewers' comments:

Reviewer's Responses to Questions

**Comments to the Author**

1. If the authors have adequately addressed your comments raised in a previous round of review and you feel that this manuscript is now acceptable for publication, you may indicate that here to bypass the “Comments to the Author” section, enter your conflict of interest statement in the “Confidential to Editor” section, and submit your "Accept" recommendation.

Reviewer #1: All comments have been addressed

Reviewer #2: All comments have been addressed

2. Is the manuscript technically sound, and do the data support the conclusions?

Reviewer #1: Yes

Reviewer #2: (No Response)

3. Has the statistical analysis been performed appropriately and rigorously? 

Reviewer #1: Yes

Reviewer #2: (No Response)

4. Have the authors made all data underlying the findings in their manuscript fully available?

Reviewer #1: No

Reviewer #2: (No Response)

5. Is the manuscript presented in an intelligible fashion and written in standard English?

Reviewer #1: Yes

Reviewer #2: (No Response)

6. Review Comments to the Author

Reviewer #1: all the comments have been addressed.

to improve the coordination between pharmacists and medical doctors, its better to understand the value of each profession by students.

Reviewer #2: (No Response)

7. PLOS authors have the option to publish the peer review history of their article (what does this mean?). If published, this will include your full peer review and any attached files.

Reviewer #1: No

Reviewer #2: No

---

## [Editor Report · Acceptance letter]

18 Aug 2021

PONE-D-21-12355R1 

How pharmacy and medicine students experience the power differential between professions: “even if the pharmacist knows better, the doctor’s decision goes” 

Dear Dr. Thomas:

I'm pleased to inform you that your manuscript has been deemed suitable for publication in PLOS ONE. Congratulations! Your manuscript is now with our production department. 

Kind regards, 

on behalf of

Dr Jenny Wilkinson 

Academic Editor

PLOS ONE